# A practical 3D-printed soft robotic prosthetic hand with multi-articulating capabilities

**Alireza Mohammadi**[1,2]*, **Jim Lavranos**[3], **Hao Zhou**[2,4], **Rahim Mutlu**[2,4], **Gursel Alici**[2,4], **Ying Tan**[1], **Peter Choong**[2,5], **Denny Oetomo**[1,2]

**1** Department of Mechanical Engineering, The University of Melbourne, Parkville, VIC, Australia, **2** Australian Research Council Centre of Excellence for Electromaterials Science, Wollongong, NSW, Australia, **3** Prosthetics and Orthotics Clinic, Caulfield Hospital, Caulfield, VIC, Australia, **4** School of Mechanical, Materials, Mechatronic and Biomedical Engineering, University of Wollongong, Wollongong, Australia, **5** Department of Surgery of University of Melbourne at St Vincent's Hospital, Fitzroy, VIC, Australia

* alirezam@unimelb.edu.au

**Data Availability Statement:** All relevant data are within the manuscript, its Supporting Information files, and a library of open-source STL files of the hand design for 3D printing, custom PCB design

## Abstract

Soft robotic hands with monolithic structure have shown great potential to be used as prostheses due to their advantages to yield light weight and compact designs as well as its ease of manufacture. However, existing soft prosthetic hands design were often not geared towards addressing some of the practical requirements highlighted in prosthetics research. The gap between the existing designs and the practical requirements significantly hampers the potential to transfer these designs to real-world applications. This work addressed these requirements with the consideration of the trade-off between practicality and performance. These requirements were achieved through exploiting the monolithic 3D printing of soft materials which incorporates membrane enclosed flexure joints in the finger designs, synergy-based thumb motion and cable-driven actuation system in the proposed hand prosthesis. Our systematic design (tentatively named X-Limb) achieves a weight of 253gr, three grasps types (with capability of individual finger movement), power-grip force of 21.5N, finger flexion speed of 1.3sec, a minimum grasping cycles of 45,000 (while maintaining its original functionality) and a bill of material cost of 200 USD (excluding quick disconnect wrist but without factoring in the cost reduction through mass production). A standard Activities Measure for Upper-Limb Amputees benchmark test was carried out to evaluate the capability of X-Limb in performing grasping task required for activities of daily living. The results show that all the practical design requirements are satisfied, and the proposed soft prosthetic hand is able to perform all the real-world grasping tasks of the benchmark tests, showing great potential in improving life quality of individuals with upper limb loss.

## Introduction

The loss of an upper limb severely affects the ability of the amputees to carry out activities of daily living (ADLs), leading to a significant impact on their well-being. An effective replacement

files of the control board, and the hand control source code are available from the GitHub repository (https://github.com/MelbourneUniHRL/X-Limb.git).

**Funding:** This project is funded by the Valma Angliss Trust and the University of Melbourne. The funders had no role in study design, data collection and analysis, decision to publish, or preparation of the manuscript.

**Competing interests:** The authors have declared that no competing interests exist.

of the lost functions by a prosthetic limb therefore has the potential to improve the quality of lives of those living with limb loss [1].

Current state-of-the-art myoelectric hand prostheses in the market such as i-limb [2], Bebionic [3] and Michelangelo [4] have mechanically rigid constructions. Due to the lack of compliance and flexibility in the structure of these robotic hand prostheses, additional mechanisms are also needed to make them safe for the physical interaction with humans and objects. For instance, passive flexible components like springs are added or using complex sensing mechanisms (such as tactile sensors) has been investigated in order to achieve the desired compliance using an extra control loop [5]. Consequently, they affect the weight of the resulting prosthetic hands and increase the complexity, need for maintenance and cost while potentially lowering durability. These factors significantly affect the user acceptability and result in a high rate of prosthetic hand abandonment of 40-60% [6, 7].

Soft robotics [8] is an emerging field which has shown great potential in addressing these issues with current hand prostheses, i.e., non-compliant structure, heavy weight and complex system. The inherent material compliance in the soft robotic systems provides safer, cheaper, and simpler mechanisms (and consequently simpler control structures) compared to the traditional rigid robotic systems [9]. Moreover, they can provide the possibility of exploiting the morphology of the structure and properties of materials to simplify the design and implementation of control strategies [10]. Therefore, in recent years, there has been a great interest in the design of prosthetic hands using soft robotic mechanisms.

Many existing soft robotic prosthetic hands are designed in laboratories with the focus of proof of concept. They can be categorised in two groups: the hands using elastic joints with cable-driven mechanisms and the hands with monolithic soft structures. ISR-SoftHand [11] and SoftHand Pro [12] are examples of the former, in the form of anthropomorphic hands with elastic joints. These hands have been shown to be highly functional. They utilise a relatively complex gearing mechanisms, resulting in a weight of more than 500gr. The latter approach fabricates the entire structure of the robotic hand with soft materials in a monolithic fashion, which eliminates a good portion of the need for assembly and the associated inadvertent misalignment between assembled parts. More importantly, it reduces the weight by removing the need for fasteners (screws, nuts and bolts) required to hold multiple segments together. Examples include RBO Hand 2 [13] and BCL-13 hand [14].

The current monolithic designs, however, utilises pneumatic actuation systems, which require air supply compressors or pumps, matrix of valves and pressure sensors. In addition to the relatively complex manufacturing process (multiple moulding procedures), the actuation system (not the hand itself) makes the whole setup bulky and heavy, which reduces its practical suitability to be used by amputees.

In this paper, the development of a practical soft robotics prosthetic hands is proposed, with a focus on the ease of manufacture and functionality for ADLs of amputees. It differs from many of the above mentioned prototypes in the incorporation of the practical requirements in the design consideration of the proposed soft robotics prosthetic hands. These requirements are based on a recent thorough survey [15], which highlighted the limitations of existing hand prostheses and the needs of upper-limb amputees, as well the practical requirements pointed out in [16]. Moreover, the practical design also takes the manufacturing process of prosthetic hands into consideration. As a summary, the following requirements need to be addressed in the design of practical soft robotic prosthetic hands:

1. Light weight (less than the average weight of a human hand) [16]

2. Intrinsic actuation system (embedded in the hand structure) [17]

3. Robust finger kinematic design and compliance in the mechanical design [15]

4. Powered thumb for multiple grasping types [15]

5. Ease of manufacture and personalisation [16]

6. Sufficient functionalities and the balance between the functionalities and complexity [15, 16]

With these practical requirements in mind, this paper presents a requirement-driven design procedure of soft robotic prosthetic hand (called X-Limb). It is of a monolithic structure as the entire hand was 3D printed from soft material, into which all the actuation and control systems are embedded. Such a novel structure addresses Requirements 1-3. The fingers of the hand are designed based on the flexure joints with a monolithic kinematic structure proposed by authors in [18] with membrane enclosure to meet Requirements 4-5. Our design also balances the needed functionalities and the complexity (Requirement 6).

The preliminary results of the proposed hand in [19] demonstrated the capability of X-Limb in grasping a wide range of objects. As an extension, this work presents and comprehensively analyses the practical properties of the X-Limb and compares its performance with other commercial and research hand prostheses. The versatility and limitations of the X-Limb are evaluated by performing the tasks required by Activities Measures of Upper-Limb Amputees (AM-ULA) benchmark test.

## Materials and methods

This section presents the detailed design requirements and the corresponding proposed solution for soft robotic prosthetic hands. Although 6 requirements are summarized from literature, when designing a prosthesis hand, such requirements are re-grouped into three categories. They are mechanical/morphological, kinodynamic, and functionality requirements.

### Design requirements

**Mechanical and morphological design.** Requirements 1-3 are related to the mechanical and morphological design, which would include appearance, size, weight, actuation system and soft robotic structure.

*Appearance*, As X-Limb is designed as a prosthetic hand, an anthropomorphic appearance and human-hand-like morphology are preferred, though the objective of the X-Limb is not to reproduce anthropomorphic operation of the human biological hand.

*Size*, The size of the X-Limb needs to be smaller than average male hand size (breadth 106±9mm and length 196±9mm) as pointed out in [20].

*Weight*, The average male hand weight is about 400gr. It was pointed out in [21] that the conventional prosthetic terminal devices weighting 400gr are considered too heavy by about 80% of myoelectric prosthetic users. The main reason is that the weight of the device is carried by the soft tissue of the amputated stump (through fitted socket and fasteners) instead of by the skeletal structure as in natural hands. Therefore, a prosthetic hand that weighs less than an average biological human hand, including all the actuation system, sensors and connectors, is preferred. A lighter prosthetic hand is particularly better for people with high-level amputation (e.g. transhumeral) because of power and weight constraints of the entire prosthetic arm [16].

*Intrinsic actuation*, Actuation configuration should be intrinsic, which means the driving, transmission, and control elements are totally embedded in the hand structure. This will provide the flexibility for prosthetists to design sockets for individuals with different level of upper-limb amputation.

*Compliance*, The whole structure of the hand which is in contact with objects and in interaction with humans should be soft/compliant.

**Kinodynamic.** Kinodynamic requirements are responding to Requirements 4-5 listed in Introduction. They consist of the grasp force and speed required for performing ADLs.

*Force*, Finding the required grasp force for performing the ADLs is challenging in general as it is largely dependent on a variety of external factors such as the friction between hand and object, the number of contact points (or surface), and the object geometry and mass [16]. Some studies in the literature suggesting 45N [22] and 68N [23] as required grasp force to carry out ADLs.

*Speed*, The speed of the prosthetic hand can be stated as finger flexion speed (degrees of flexion per sec) or the total time to open or close the hand. According to the literature, the sufficient prosthetic hand closing speed for conducting ADLs ranges from 0.8 sec [24] to 1.5 sec [25]. A very high speed is not required for prosthetic hands as most of myoelectric control algorithms rely on the user to stop the hand at the right closing position; thus high hand closing speed will make it more difficult [16].

One of the prosthetic hand components which significantly affects the grasp force and speed is the cosmetic glove. Cosmetic gloves are the common approach in current commercial hand prostheses to protect the prosthetic hand from dirt/moisture and provide cosmetic appearance. However, different gloves have different values of stiffness so the grip force/speed of the hand will vary depending on the glove, and this effect cannot be taken into account in the design of the hand. In addition, difficulty to put on such gloves, glove durability and the cost of glove replacement are among the main reasons for prosthesis dissatisfaction [15]. In the proposed soft prosthetic hand, this effect should be considered in design of the hand.

**Functionality.** Corresponding to Requirement 6, the functionality and dexterity in performing ADLs by amputees play important roles in the design of prosthetic hands. Higher functionalities generally mean higher complexity in the design of the mechanism. In order to address such a trade-off, the major functionalities of prosthetic hands need to be identified from benchmark tests.

This work utilizes the Activities Measure for Upper-Limb Amputees (AM-ULA), which is one of the most common benchmark tests [26], to identify key functionalities needed for a prosthetic hand. Such a measure takes a variety of aspects of activity performance into consideration. It includes: sub-task completion, skilfulness of prosthesis use, movement quality, independence, and overall time to perform that activity. The test has 18 items as listed in Table 1, each of which is scored from 0–4 (unable to excellent), with higher scores indicating better functional performance.

Table 1 contains 11 bimanual and 7 monomanual activities. Some of the bimanual activities can be performed with one hand (e.g. 'Put on socks'), especially for subjects with amputation on their non-dominant side. Some of the activities are extremely challenging or impossible to perform with one hand (e.g. 'Attach end of zipper and zip' and 'Tie shoe laces') and these are the items for which subjects more frequently use their prosthesis [26]. In addition, the activities can be completed in more than one way. For instance, to use hammer and nail, the subject can hold the nail with prosthetic hand and use the intact hand for holding hammer or the other way around. The detailed definition of the tasks and instructions to guide the subjects in performing the tasks are provided in [26]. The last column of Table 1 shows the grasp primitives which enable the hand to perform the corresponding tasks. In performing power grasp, all fingers will move together towards the centre of the palm. In performing pinch grasp, the thumb will move to abduction position and then the index finger will move to meet the thumb. In tripod grip, when the thumb is opposed, the index and the middle fingers will move to meet the thumb. It can therefore be observed that the three grasps will span all the activities

**Table 1. AM-ULA items [26] and the percentage the activity is performed using the prosthetic hand with unilateral amputees [27].**

| # | Task | Using prosthesis (%) | Required grasp type |
|---|------|---------------------|---------------------|
| 1 | Attach end of zipper and zip | 63.6 | Pinch |
| 2 | Tie shoe laces | 50.0 | Pinch |
| 3 | Fold a bath towel | 43.1 | Pinch |
| 4 | Use a hammer and nail | 42.2 | Power/Pinch |
| 5 | Use scissors | 35.0 | Tripod |
| 6 | Button shirt | 16.7 | Pinch/Tripod |
| 7 | Put on T-shirt | 11.5 | Pinch/Tripod |
| 8 | Remove T-shirt | 11.5 | Pinch/Tripod |
| 9 | Use phone | 6.8 | Power |
| 10 | Use fork | 6.6 | Power |
| 11 | Use spoon | 6.6 | Power |
| 12 | Write name legibly | 6.6 | Tripod |
| 13 | Drink from a paper cup | 5.1 | Power |
| 14 | Put on socks | 3.4 | Power |
| 15 | Brush/comb hair | 1.6 | Power |
| 16 | Pour soda/water | - | Power |
| 17 | Door knob | - | Power |
| 18 | Reach overhead | - | Power |

identified in AM-ULA. Hence, the pinch, tripod and power grasps are the focus in the prosthetic hand design.

## Proposed solutions

In this section, the proposed solutions are presented to address the aforementioned requirements (see the summary in Table 2). Each solution is detailed in the following subsections respectively.

**3D-Printing-based fabrication.** The X-Limb is specifically designed for fabrication using 3D printing of soft materials due to the capability of 3D printing methods in delivering very sophisticated and complex geometries with no need of post-processing. In addition, 3D printing allows the fabrication of customised products in low volumes in a cost-effective way which is of interest in the fabrication of hand prostheses.

The 3D printing method that we used is the Fused Deposition Modeling (FDM) technique which is one of the well-established methods. With the FDM method, it is possible to have enclosed hollow spaces in the structure of the hand and the size of these spaces can be adjusted using the infill percentage in the 3D printer software. With this method, the weight of the

**Table 2. Proposed solutions for different design requirements.**

| Design requirement | Proposed Solutions |
|--------------------|--------------------|
| Mechanical/Morphological | 3D-printing of soft materials |
| Mechanical/Morphological | Fingers with flexure joints |
| Kinodynamic and Morphological | Membrane enclosure |
| Functionality | Thumb design |
| Functionality | Fingertip design |
| Kinodynamic and Functionality | Actuation system |

hand can be reduced significantly. Importantly, membrane enclosed flexure joints can be produced monolithically (see next subsection for details).

The flexible and soft material which is used for fabrication of the X-Limb is TPU90 (Thermoplastic Polyurethane with Shore 90A). In addition to its high elasticity and compliance [28], TPU has other advantages such as ease of synthesis, low cost and biocompatibility [29]. For the 3D printing of TPU, the commercially available 3D printer (FlashForge Guider II) was used. The FlashForge Guider II has a large printing volume which enables printing the whole hand as a single print. In the case that such a large 3D printer is not available, the fingers and palm of the hand can be printed separately and then fused together. This method is also explained in S2 Video using a 3D printer with small printing volume (FlashForge Dreamer).

The 3D printer used in this paper (FlashForge) has an open-source software (FlashPrint) for slicing the STL files and preparing the file for 3D printing. In this software, by choosing the printing material, the extrusion speed and travel speed will be set automatically to an appropriate values. In our case "Flexible Filament" is used as Material Type which results in the extrusion speed of $60 mm/s$ and travel speed of $80 mm/s$. The nozzle temperature was set to $220°C$ which is the recommended value by TPU filament (FilaForm) manufacturer, considering the glass transition temperature and melting temperature of TPU at -44°C and 195°C, respectively [30].

It should be noted that the optimal infill percentage depends on the material used for the 3D printing and also the role of the 3D printed structure. The infill density of the 3D printed parts affects the stiffness of the flexure joints. Therefore, we need to choose the infill percentage of the body of the fingers of the hand such that it provides enough strength in the finger joints while not significantly increasing the stiffness of these joints, as that would increase the force required for finger flexion [18]. We used 30% infill for fingers and 20% infill for palm of the hand. The infill pattern was hexagon.

**Fingers with flexure joints.** The fingers are designed using flexure joints (hinges) with a monolithic structure as shown in Fig 1. The flexure joints considered in the scope of this paper are those constructed out of compliant material [18, 31], thus capable of producing gross displacement. These joints may not perform an accurate angular displacement about a single (constant) axis of rotation.

The morphology of the flexure joints has a significant effect on their stiffness. The effect of morphology has been studied in detail in [18] for the flexure joints. In this paper, the most common geometry in designing the single-axis compliant flexure joints is used, which is a corner-filleted groove, as shown in Fig 1(a), where the variable $t$ is the thickness of the flexure at its thinnest point, $c$ is the depth, $r$ is the fillet radius and $L$ is the length of the flexure. Fig 1(b) shows a soft monolithic fingers constructed based on these corner-filleted flexure joints.

The relative flexural stiffness of the finger joints affect the kinematics and dynamics of the hand fingers. Depending on the stiffness of the finger joints, the trajectory of the finger joints will be different. The trajectory of the fingertip is especially important in precision grasping. In [32], a systematic procedure is proposed to compute the relative stiffness of the compliant joints to obtain a desired trajectory for the fingertip.

**Membrane enclosure.** These compliant fingers with flexure joints are elastic and can bend, twist, stretch, compress, buckle, and wrinkle, to various extents [9]. The flexural bending of the joints are designed and intentional, while the other deformation of the structure are to be minimised. The flexural degrees of freedom of the fingers are explicitly actuated. These fingers are often underactuated and contain some dependent DoFs. Cable-driven mechanisms are among the most common actuation methods for these underactuated soft flexure joints.

There are two main issues in practical implementation of the cable-driven robotic hands. One is that the cable exposed at the flexure joints (See Fig 1(b)) carries the risk of being caught

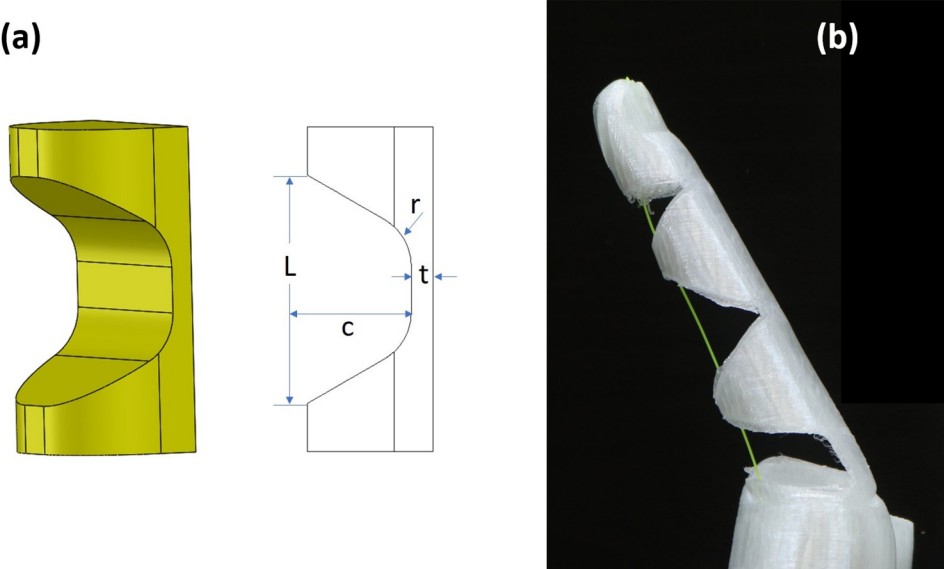

**Fig 1. A soft monolithic finger with flexure joints: (a) Corner-filleted flexure joint and design parameters; (b) Finger with three corner-filleted flexure joints (one actuation cable is threaded through the 3 flexure joints).**

while grasping objects. The other is that the foreign objects can become lodged in the joint or enter the mechanism through the grove meant for the actuation cable. The common approach to overcome the issue is to use a glove to cover the hand mechanism, which often still does not satisfactorily address the practical needs as pointed out in requirements [15].

In [19], the idea of constructing a membrane as part of the monolithic structure of the mechanism to cover the flexure joint (Fig 2) was introduced by the authors. The membrane,

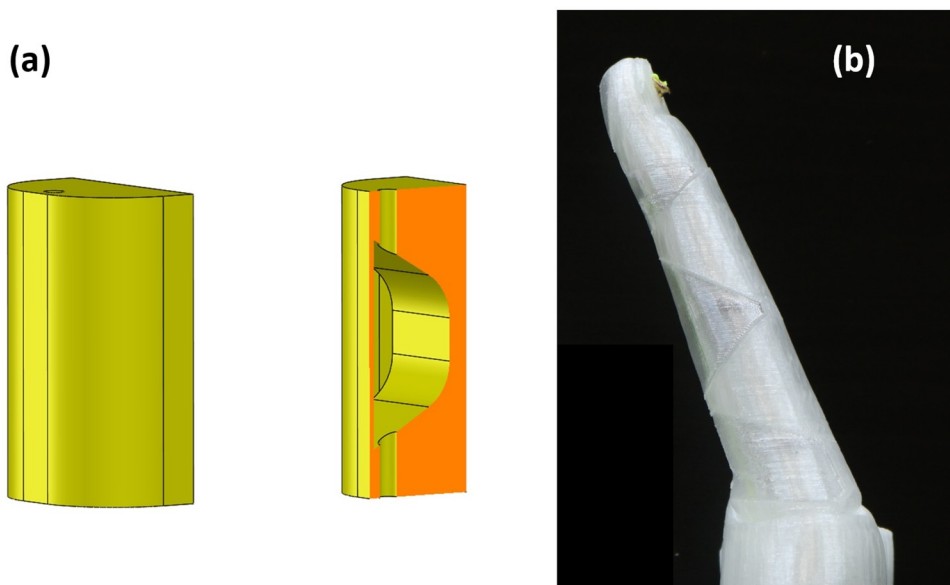

**Fig 2. A soft monolithic finger with membrane enclosure: (a) Corner-filleted flexure joint with membrane enclosure; (b) Finger with three membrane enclosed flexure joints.**

printed in the same material as the hand structure, encloses the entire finger and its actuation cable. This prevents the cables from being caught on external objects during grasping and enables more streamlined design similar to human fingers. Moreover, it provides the possibility of potentially water-proof and dust-proof design. In S1 Video, it is shown how to thread the tendon cable through the flexure joints with membrane enclosure.

In order to construct a membrane, a thin layer is added over the flexure joint in CAD design as shown in Fig 2(a). To fabricate the flexure joints with the membrane, 3D printing techniques are employed. The main reason is that the other soft material fabrication techniques such as Shape Deposition Manufacturing (SDM) [33] are not capable of producing a hollow space that is completely enclosed. It should be noted that the joint must be printed without any support material otherwise it will fill in the void between the flexure joint and the membrane. Being completely enclosed, the support material cannot be removed post printing. Since the flexure joint has an angle relative to the vertical, it is possible to print a layer of material with the previous layer acting as its support, thus removing the need for support material.

The addition of the membrane changes the mechanical properties of the compliant flexure joints including torsional and flexural stiffness. Torsional stiffness refers to the torque required to twist the joint. The torsional stiffness is mainly important in preventing out-of-plane motion of the fingers. The reduction of out-of-plane motion is critical in enabling stable fingertip grasps (precision pinch grasp). Flexural stiffness refers to the resistance of the joint to the bending moment in the desired direction of motion. Later, the effect of membrane enclosure on flexure joints stiffness will be tested.

**Thumb design.** Four fingers of the hand (index, middle, ring and little) are constructed based on the design shown in Fig 2. The remaining finger is the thumb. The thumb of a prosthetic hand plays a significant role in realising different grasp types, much like the important role of the opposable thumb in the biological human hand. Depending on the preshape of the thumb, prosthetic hand users are able to perform different grasps such as pinch, tripod and power identified from AM-ULA.

In order to realise the three grasp types (pinch, tripod and power), at least two degrees of freedom (DoF) are needed for the thumb, one for adduction/abduction of the thumb and the other for flexion/extension. However, independent actuation of these two DoFs of the thumb will increase the complexity of the hand design. For this reason, some prosthetic hands such as SensorHand from OttoBock have only the abduction preshape of the thumb to simplify the design, which limits performance / functionality, such as having only a single grasp type, small grip size (grip opening) and an unnatural pose of the hand.

In current state-of-the-art commercial hand prostheses, two actuators are used to realise the required grasp types, one for adduction/abduction of the thumb and one for flexion/extension. In some of these hands, only one active actuator is used for flexion/extension of the thumb and the other DoF is manipulated manually (e.g. Bebionic from RSL Steeper Inc.).

In [34], the authors proposed the actuation of both DoFs using one actuator through using the kinematic synergy of the two degrees of freedom. This paper modifies the proposed design in [34] through the integration of the membrane enclosure in the thumb design and removing the distal and proximal joints to provide stable and strong pinch and power grasps. Moreover, the axis of rotation of the thumb is adjusted in such a way that the actuation of the thumb will move the thumb from the natural position towards the center of the palm providing power grasp (as shown in Fig 3) and stops in a position that with meet the index and middle fingers (if actuated) to yield the pinch/tripod grasp. Such a design removes the need for extra command signal for the preshaping of the thumb for pinch and power grasps and simplifies the overall structure of the hand.

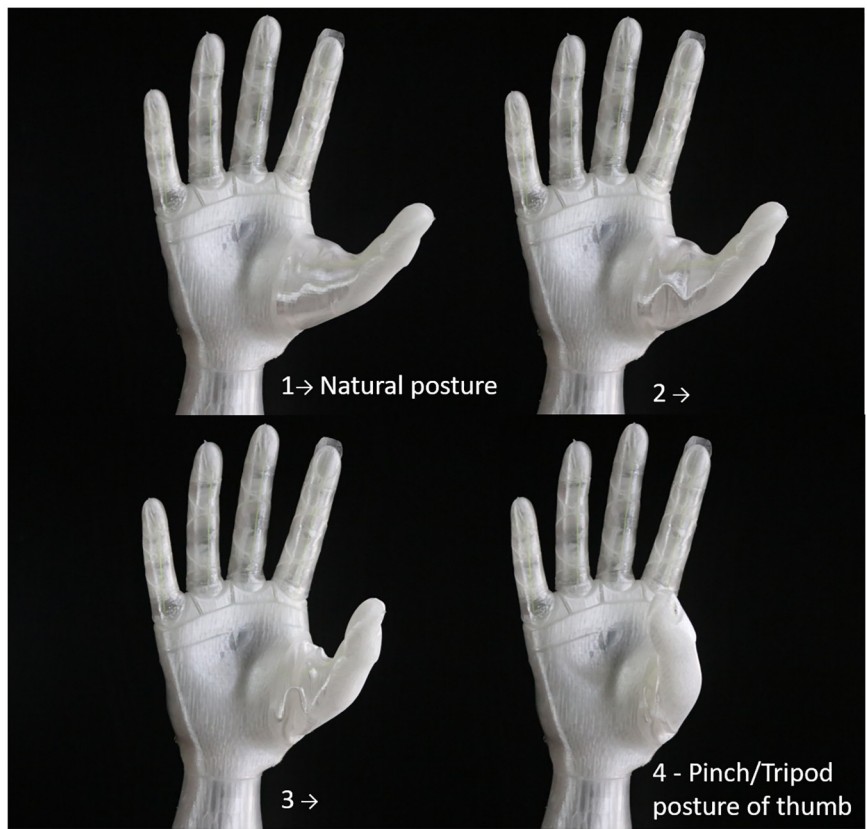

**Fig 3. Articulation of thumb from natural position towards pinch/tripod position of the thumb.** While moving from position 1 to 4, in coordination with other fingers, it is able to perform power grasp.

**Fingertip design.** In the fingertip design, three features are considered for a stable and robustly precise pinch grasp while keeping the mechanism simple at no extra cost or complication of the hand fabrication.

The first feature is the addition of fingernail in the index finger as shown in Fig 4(a). This feature helps in picking up and holding small and thin objects. The fingernails are 3D printed along with the whole structure of the finger so they do not require extra assembly. They are

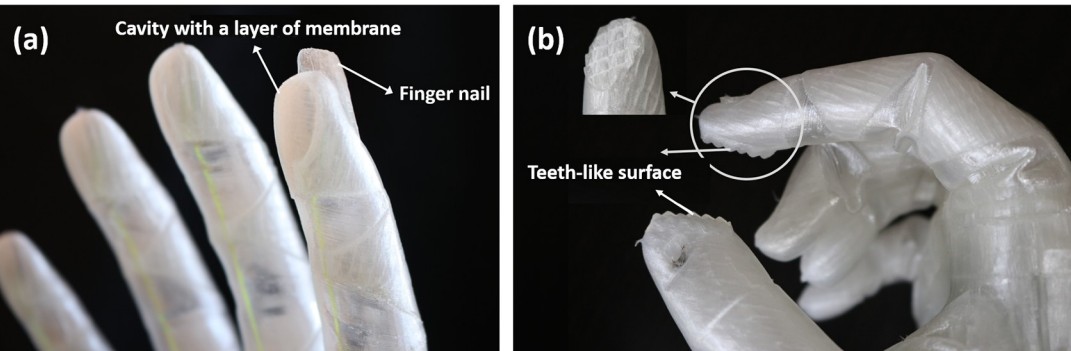

**Fig 4. Fingertip build-in features: (a) 3D printed finger nail for collecting thin objects such as coins and a cavity in the fingertip for increasing the contact surface area in pinch grasp; (b) 3D printed teeth-like surface on index fingertip and lateral side of the thumb for stable pinch grasp.**

also compliant and therefore safe in contact with human body and environment. The second feature is a cavity with a layer of membrane 3D printed in fingertip of the index finger, as shown in Fig 4(a). This feature increases the contact surface between the fingertip and object being grasped in pinch grasp. The third feature considers uneven surface such as teeth-like surface as shown in Fig 4(b). Such a surface can be 3D printed along with the finger to increase the contact surface area and provide more robust and stable pinch grip for tasks such as doing shoe laces with lower amount of grip force.

**Actuation and control system.**   Considering the functionality requirements of the grasp types needed, three independently controlled movement of the fingers are required to realise the required grasps (power and pinch/tripod): the thumb, the index, and the other three fingers. Since all the motors, controller and position/current sensors need to be embedded inside the hand, the main limitation in the selection of the actuators is the physical space. In addition, the selected motors should satisfy the kinodynamic requirements for grasp force and speed.

In the X-Limb design, each of index, middle, ring, and little fingers has three joints (distal, proximal, and metacarpal) as shown in Figs 1 and 2 and thumb has one metacarpal joint as shown in Fig 3 which results in 13 DoFs of the X-Limb. One actuator is required to actuate each of these underactuated fingers.

Based on the size of a medium male hand, it is possible to fit within the hand the actuation system that consists of five geared DC micromotors (6V HPCB Micro Metal Gearmotor, Pololu Inc.), see Fig 5. Each of these motors actuates one of the fingers. Although we can use one large DC motor for co-actuation of middle, ring and little fingers, due to limited space, it cannot fit in the palm of the hand. Using one motor to actuate three fingers with different phalangeal lengths also required a mechanical transmission system to divide the torque of the

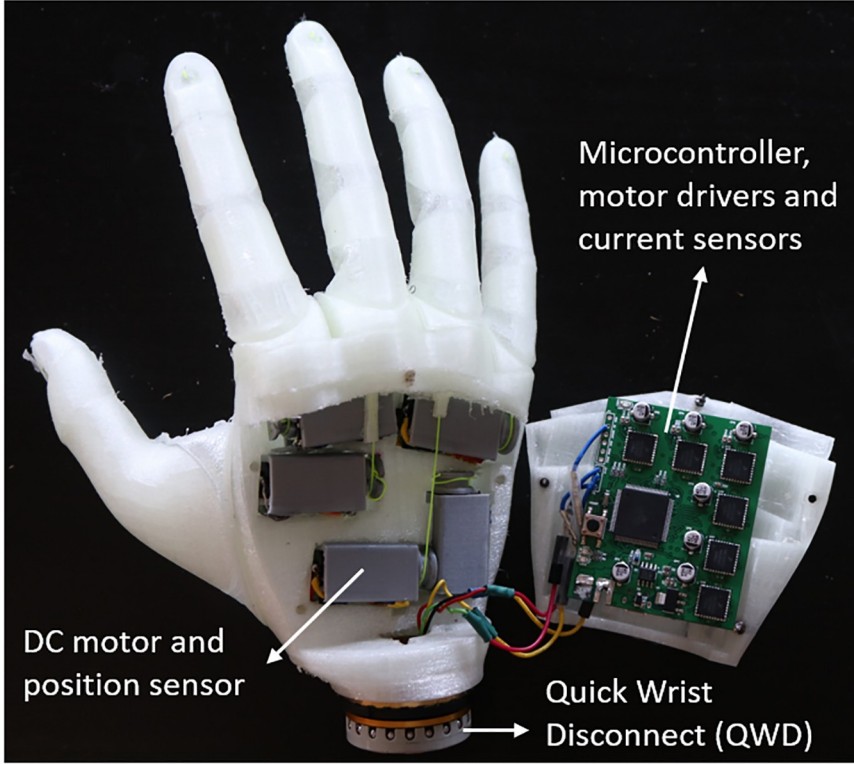

**Fig 5. X-Limb actuation mechanism and control system.**

motor among the fingers, such as shown in [35]. Such mechanical transmission also require space that is not easily available within the hand. By using individual actuation for each of fingers, the required power is divided among the three motors. In addition, it provides the ability to individually actuate fingers, allowing more complex grasps to be realised if needed.

A cable-driven mechanism is employed as the actuation transmission to the fingers. Tendon cables for each finger are wrapped around small spools and connected to the motors. The detailed assembly of actuation and control system is provided in S3 Video. The diameter of these spools are designed to an appropriate trade-off between the actuation force and speed. Increasing the diameter of the spool results in lower pulling force but faster flexion of the finger. A potentiometer is attached to the shaft of each motor to measure the position of the shaft. Actuation configuration is intrinsic and all the actuators, motor drivers and the controller are embedded in the hand structure as shown in Fig 5.

All five motors are controlled with a custom designed PCB with microcontroller board which is based on Atmel 8-bit ATmega2560 and Freescale MC33926 H-bridge motor driver with embedded current sensor. The motor driver can supply up to almost 3A continuous current to one brushed DC motor at 6V, and it can tolerate peak currents up to 5A per channel for a few seconds. The DC motors are 9.5gr with no load speed of 30,000rpm and stall torque of 1Nmm, which can be fitted with different gearheads of different ratios. For instance, with gearhead of 298:1 it can provide stall torque of 0.34N and no-load speed of 110rpm. Considering the required cable length excursion of different fingers, required full flexion speed of fingers and available space inside the palm, the gearhead ratio of 1000:1 is used for thumb, index, middle and ring fingers, ratio of 298:1 for little finger, spool with diameter of 25mm for middle and ring fingers and 18mm for thumb, index and little fingers.

It should be noted that the motors are only pulling the tendon cable for flexion of the finger and the springlike compliance in the flexure joint provides the force for extension of the fingers. This significantly simplifies the finger mechanism and number of actuators required for control of the hand resulting in fewer components for assembly and in turn, fewer potential maintenance issues.

The hand is controlled by using only two sEMG (surface electromyography) signals provided by most of commercial EMG electrodes for opening and closing the hand. A clinical grade quick disconnect wrist (QDW) is used for the quick mounting and release of the X-Limb to the existing socket of the myoelectric hand users (Fig 5). One tactile button located below the thumb towards the back of the hand to scroll through different grips rather than using the EMG signals (holding the hand open command for certain amount of time) as per current practice in the commercial hand prostheses. The button can be operated with one hand by hitting it against a surface (test subjects have occasionally hit it against their hip out of convenience). The experimental results in [36] demonstrated that first time subjects were more effective in performing the task of selecting the right preshape using the buttons than through the attempt of producing the right EMG signals (as seen in some commercial devices). This has the potential to significantly reduce the challenges in learning to operate a myoelectric hand prosthesis.

## Results and discussion

This section evaluates the practical performance of the designed X-Limb using various experimental settings. Comparisons of the proposed X-Limb with available commercial products and laboratory prototypes are also presented. Furthermore, the durability of the X-Limb, which plays an important role in practical consideration of prosthetic devices, is tested and presented.

## Mechanical and morphological characteristics

Fig 6 shows the time lapse of 3D printing of the X-Limb. It takes 34 hours to print the palm and fingers and 4 hours for printing of the back cover. More details are provided in the Supplementary Materials.

As shown in Fig 7, the designed soft robotic prosthetic hand has an anthropomorphic appearance and human-hand-like morphology. The size of the final design is a hand with breadth of 92mm and length of 191mm which is smaller than an average male hand size (breadth 106±9mm and length 196±9mm) and slightly larger than an average female hand size (breadth 91±5mm and length 179±10mm) [20].

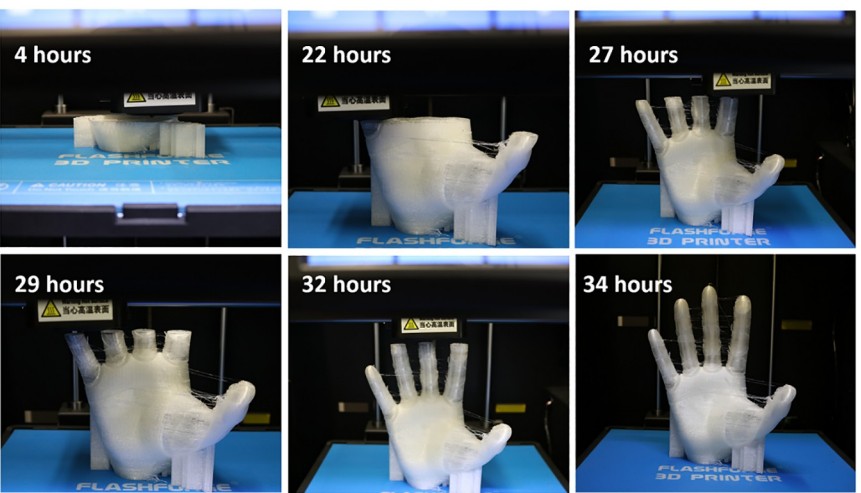

**Fig 6. Time lapse of 3D printing of X-Limb.**

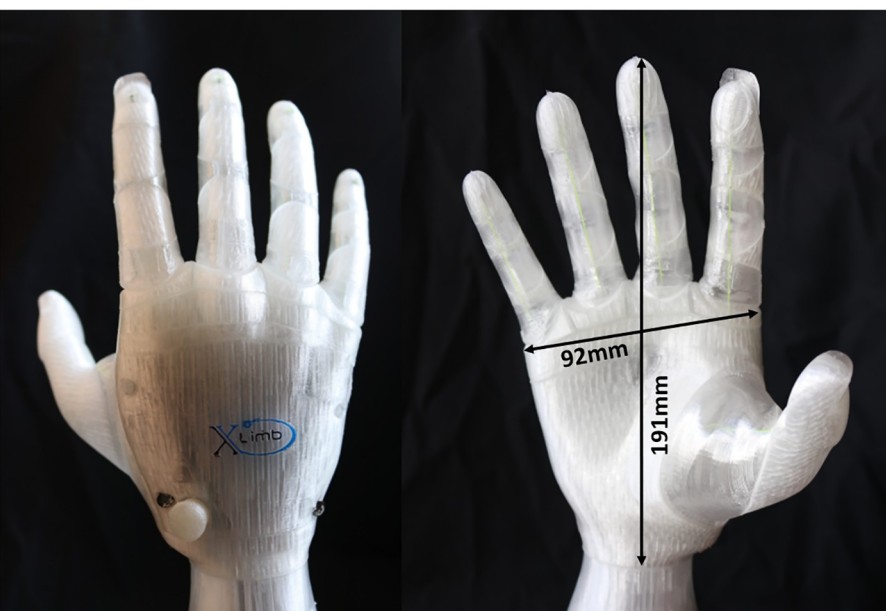

**Fig 7. X-Limb appearance and size.**

**Table 3. Characteristics of the proposed design (X-Limb) and other anthropomorphic commercial and research myoelectric hand prostheses with intrinsic actuation [16].**

|  | Weight (gr) | # DoF | # Actuators | Grip force (N) [Pinch,Power] | Grasp speed |
|---|---|---|---|---|---|
| **Commercial** |  |  |  |  |  |
| SensorHand (OttoBock) | 350-500 | 1 | 1 | [-,100] | 300mm/s at tip |
| i-limb Ultra Revolution (Touch Bionics) [2] | 504 | 6 | 5 | [6.54,136] | 1.2 sec |
| Bebionic (RSL Steeper) | 495-539 | 6 | 5 | [12.47,77] | 1.9 sec |
| Michelangelo (OttoBock) | 420 | 2 | 2 | [-,80] | - |
| **Research** |  |  |  |  |  |
| Remedi | 400 | 6 | 6 | [-,9.2] | - |
| MANUS Hand | 1200 | 3 | 2 | [,60] | 2.5 sec |
| Smart Hand | 520 | 16 | 4 | [-,18] | 1.4 sec |
| Fluid Hand III | 400 | 8 | 1 pump | [-,45] | 1 sec |
| SoftHand Pro | 520 | 2 | 2 | [20, 40] | 1.5 sec |
| **X-Limb** | 253 | 13 | 5 | [10.2,21.5] | 1.3 sec |

The overall weight of the X-Limb hand including the embedded sensors, actuators, controller, using a digital scale, is 253gr which is 36% less than an average male hand weight of 400gr [21]. In the cases where the quick disconnect wrist (QDW) is needed, the weight of the hand including QDW is 313gr. This weight is potentially the lowest amongst all the commercial and research hands with intrinsic actuation system at this current time, as shown in Table 3.

## Kinodynamic characteristics

The grasping speed and force are tested against the kinodynamic requirements.

**Speed**. The hand grasp speed is defined as the finger flexion speed or the time required for full flexion of the hand. For the X-Limb, the time required for motor spool to pull the tendon cable from full extension to full flexion of the middle finger (as the slowest finger) is obtained using the position sensor attache to the motor shaft. The obtained time is 1.3sec which satisfies the design requirement for the hand closing speed. In addition, by changing the size of DC motor spool or changing gear ratio of DC motor gearhead, we are able to adjust the trade-off between grip strength and grip speed based on the requirements and priorities. For instance, using gearhead ratio of 298:1 instead of 1000:1 will result in hand closing speed of 0.9 sec and pinch/power grip force of 4.5/11N [19]. The comparisons of hand closing speed between the X-Limb and all the commercial and research hands are listed in Table 3 and shows that the X-Limb has a comparable speed.

**Force**. In order to measure the pinch grip forces, a force-sensitive resistor (FSR) sensor is placed between two 3D printed plates with a circular knob to provide constant contact surface on FSR sensor. The sensor is calibrated with different weights and then used to measure the pinch force grip as shown in Fig 8(a). The maximum pinch grip force (before the index finger loses contact with the thumb) was measured to be 10.2±0.4N at 50 iterations of the force measurement. The power grip force was measured using a cylindrical object with diameter of 40mm and two FSR sensors placed in the middle (Fig 8(b)). This experiment is also iterated for 50 times which resulted in power grip force of 21.5±1.2N at full flexion of the fingers.

Comparison of X-Limb grip force with other commercial and research hands is shown in Fig 8(c) and 8(d). The results show that, considering the payload to weight ratio, the grip force characteristics of X-Limb is similar to the SoftHand Pro (see Table 3) and its ratio is higher than other research hands and lower than commercial hand prostheses. The high grip force of

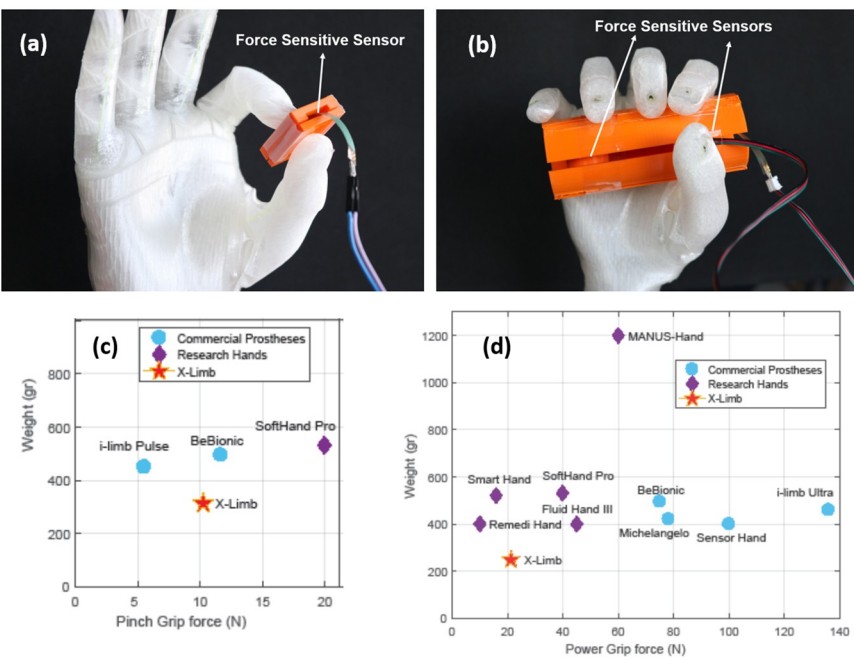

**Fig 8.** Power and pinch grip force measurement: (a) Measurement setup of pinch grip force; (b) Measurement setup of power grip force. Distribution of hand weight compared with grip force of existing commercial and research hand prostheses for: (c) pinch grasp and (d) power grasp.

commercial hand prostheses, as reported in [37], is due to their non-adaptive configuration. They exert high grip forces that are concentrated on a small contact area while adaptive prosthetic hands conform to the shape of objects and exert contact forces which are distributed over a wide contact area. The force distribution of adaptive hand prostheses is similar to that of a natural human hand. This is one of the main reasons that hand prostheses with compliant structure are safer for interaction with objects without the need for an explicit tactile feedback or complex control algorithms.

The fingertip features of the X-Limb, as discussed in the previous section, also contributed to the performance in the ADLs with lower amounts of force. One example of the tasks that require high pinch force is tying up the shoe laces, which can be managed with fingertip teeth of index and thumb as shown in Fig 15(d). The increase in the friction in holding the shoe lace means that a lower pinch force is required.

It should be noted that the force exerted on an object is a function of hand posture, contact surface and object geometry/diameter and can be lower or higher than the maximum grip forces reported here. Furthermore, it should be pointed out that the effect of cosmetic gloves is not considered in the reported grasp force of research and commercial hand prostheses while the membrane enclosure effect (which is considered as glove for X-Limb) has been taken into account in the grasp force and analysed in detail in the following subsection. The prosthetic cosmetic gloves add a significant amount of stiffness to the hand and therefore would introduce a non-negligible effect on the grasp force of the prosthesis.

**Stiffness of flexure joints with membrane enclosure.** As shown in Table 2, the stiffness of flexure joints with membrane enclosure is related to both the morphological and kinodynamic characteristics. The membrane enclosure protects the joints and actuation cable from external interference and provides the cosmetic appearance. At the same time, it increases the stiffness of the hand (fingers) which increases the motor actuation torque required about the

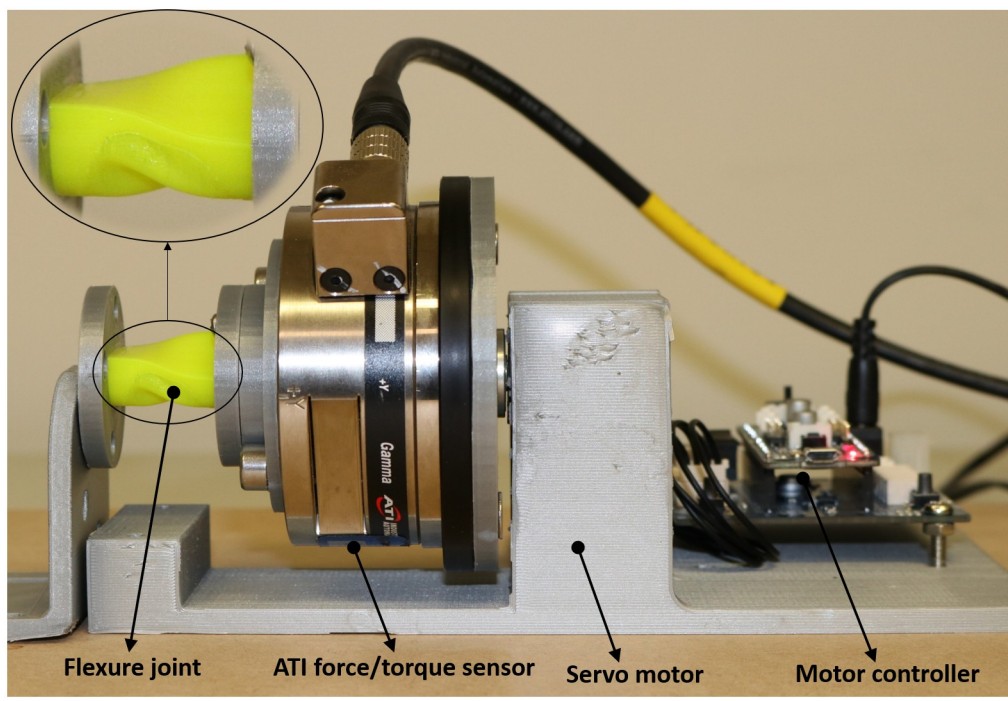

**Fig 9. Experimental setup for measuring torsional stiffness.**

flexural degrees of freedom while improving the structural rigidity of the fingers in the directions which are meant to be rigid. As a result, appropriate experiments are designed to test the stiffness of the flexure joints with membrane enclosure.

The experimental setup for measuring the torsional stiffness is shown in Fig 9. The torsional deformation is the one that benefits from the improved rigidity introduced by the membrane around the flexure joints, where the higher torsional stiffness decreases the undesired out-of-plane motion in the directions generally assumed to be well constrained. The experimental setup consists of a Dynamixel servo motor (XM430-W210), ATI 6-Axis Force/Torque sensor and Robotis OpenCM9.04 microcontroller for control of the servo motor. The servo motor is used to actuate the finger which consists of only one flexure joint. In order to measure the torsional stiffness, the ATI sensor is connected to one end of the finger joint while the other end of that finger is fixed. The finger is actuated in 2 degree increments and the quasi-static torque required for the case of the joint with membrane and without membrane were measured. The experiment carried out for the flexure joint without membrane and with 0.5mm membrane and 3D printed with 30% infill. Dimensions of the flexure joint are $L = 20mm$, $r = 5mm$, $t = 2mm$ and $c = 9.5mm$ (See Fig 1(a)).

Fig 9. Experimental setup for measuring torsional stiffness. The result of torsional stiffness test is shown in Fig 10. It shows a significant increase in the torque needed to produce the incremental displacement (the torsional stiffness) of the joint, approximately doubled that needed for the joint without membrane (for the given flexure geometry and membrane thickness).

To evaluate the flexural stiffness of the joint using cable actuation, the tendon cable is attached to Mecmesin force testing machine (Mecmesin Ltd, England) (Fig 11). The tendon cable force is measured using the load cell attached to the force testing machine. The sequential images of the joint flexion has been processed in SolidWorks (Dassault Systems Inc.) to obtain

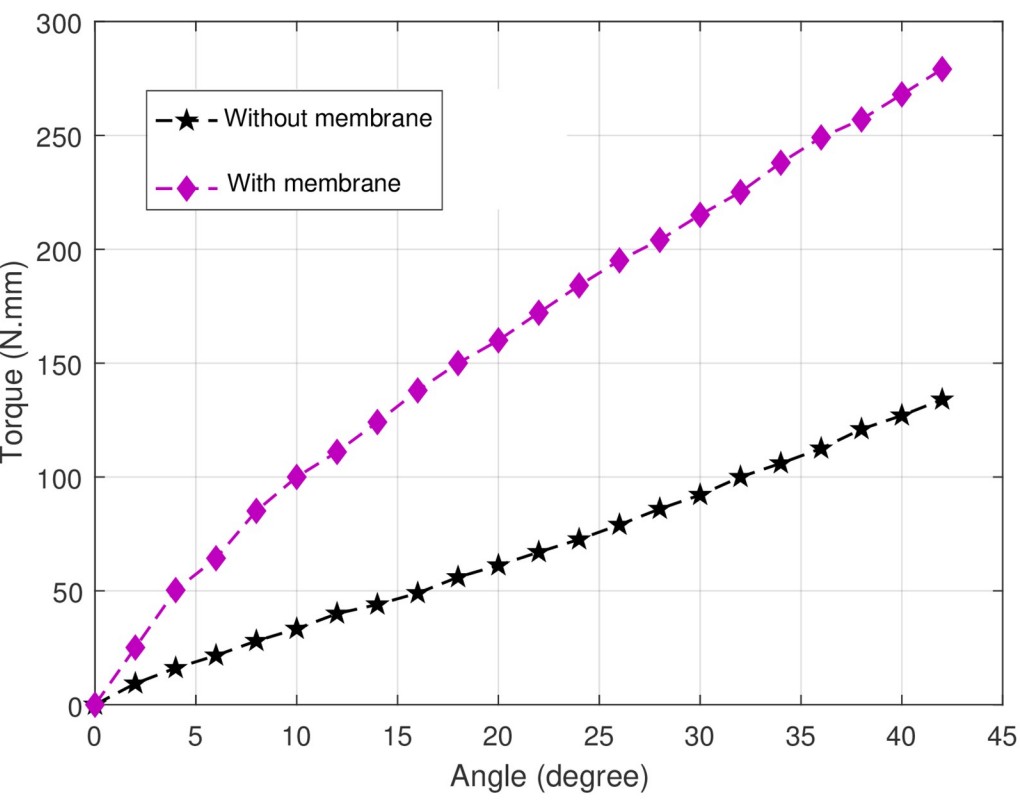

**Fig 10. Experimental results of torsional stiffness of flexure joints with and without membrane.**

the flexion angle at each applied force measurement. The experiment carried out for the flexure joint without membrane and with 0.5mm membrane.

The results of flexural stiffness in Fig 12 show that the membrane adds stiffness to the flexural degree of freedom in a non-linear fashion. There is an initial high stiffness when the fingers are fully extended, requiring a large actuation cable force to initiate bending. This force reduces significantly once the bend is initiated as the membrane buckles and forms a fold. The

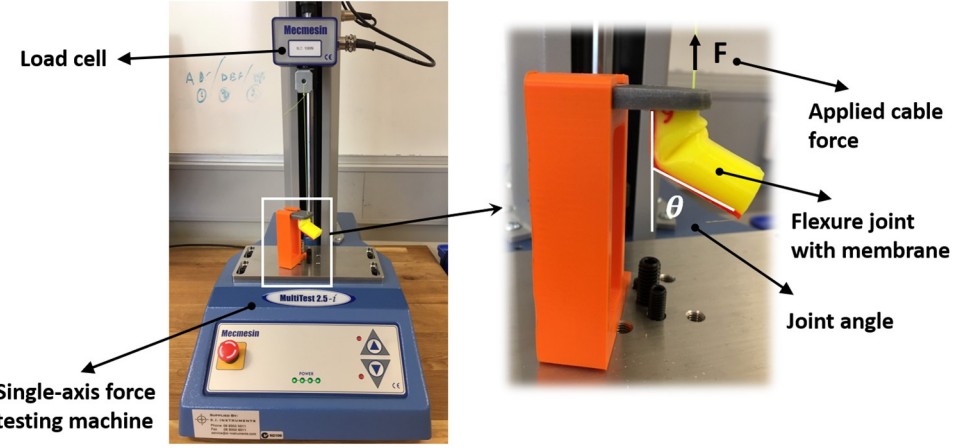

**Fig 11. Experimental setup for measuring flexural stiffness.**

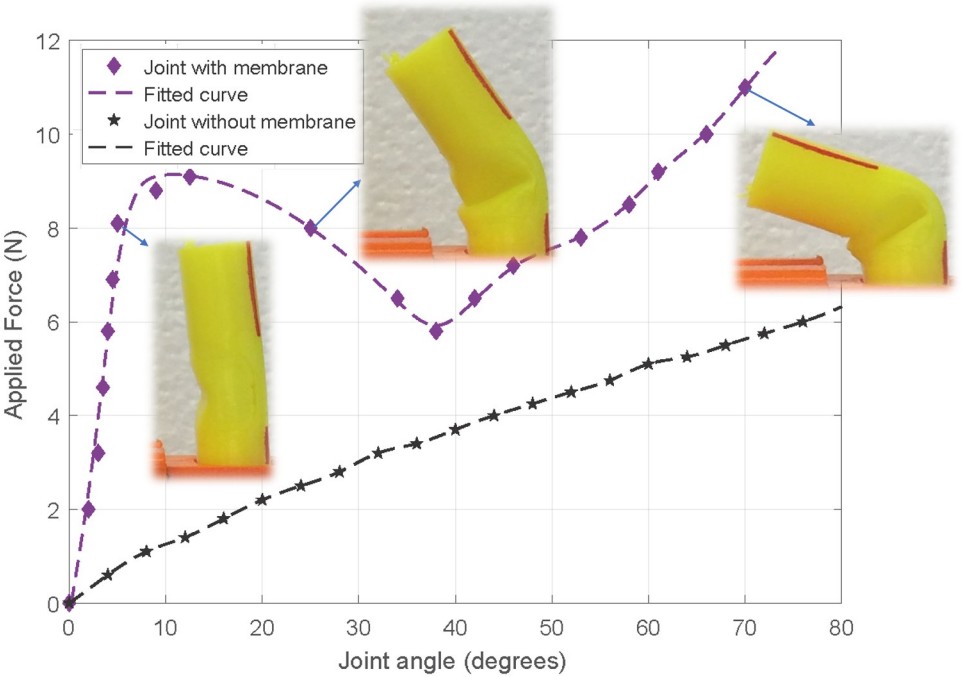

**Fig 12. Experimental results of flexural stiffness of the flexure joints with and without membrane.**

stiffness then increases almost linearly from then on. Overall flexural stiffness of the joint is increased with the membrane. As a result, higher actuation force is required for deflection of the joint. On the other hand, in the cases that the joints do not have extensor tendons, the joint elasticity alone brings the joint to the original straight position by relaxing the flexor tendon wire. Therefore, the flexure joint should have enough flexural stiffness to return it to its original position. This highlights the importance of choosing an optimal flexural stiffness in design of the fingers constructed by flexure joints. For instance, since a minimum amount of flexure joint stiffness is required to return the finger to its original position in joints without membrane, we can use a thinner flexure joint than the minimum stiffness required and instead use the membrane to provide the required flexural stiffness. As a result, with optimal selection of flexure joint thickness, we can have all the advantages of the membrane without the need for higher actuation force.

## Functionality

In this subsection, performance of the X-Limb in carrying out the tasks of AM-ULA benchmark test (as listed in Table 1) is demonstrated. A combination of position-control and torque-control has been used for performing different grasping tasks. In each grasp type, the corresponding fingers of the hand will move with different speed to the predefined position while receiving the EMG signal from user. For instance, as shown in Fig 13, moving of thumb from adduction to abduction position in coordination with other fingers results in the pinch grip. The middle, ring and little fingers are flexed with maximum speed while the index finger will move at slower speed so while the thumb is in the abduction position, the index finger will continue the flexion for pinch grasping. The finger flexion and extension will stop either when user stops sending the EMG signal (stop contracting their muscle) or when the predefined

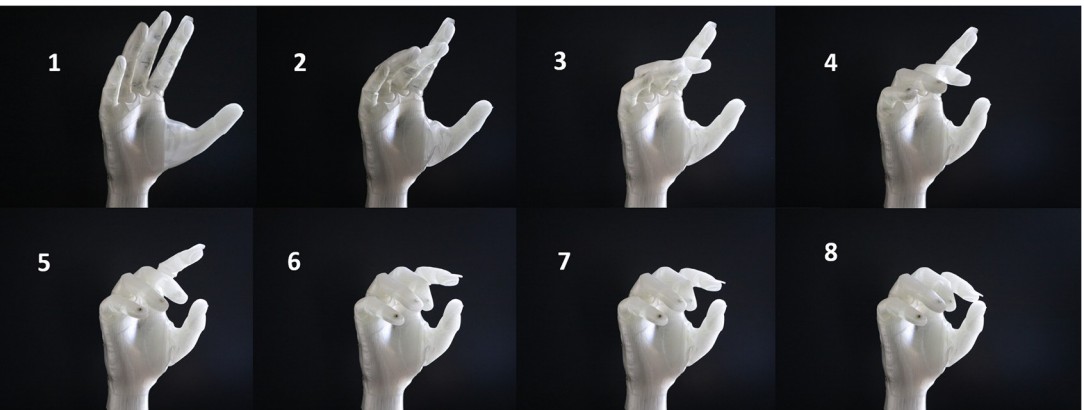

**Fig 13. Synergy of fingers for performing pinch grasp.**

threshold of the motor torque is achieved. Therefore, when fingers of the hand are in contact with an object, further flexion of the fingers will result in higher force exertion on the object.

As shown in Fig 14 and S4 Video, different design solutions of the X-Limb enables performing the grasp types required for carrying out the tasks of the AM-ULA benchmark test as listed in Table 1.

In order to explain further capabilities of the X-Limb, which are not required by AM-ULA, a few more grasping tasks are performed. As shown in Fig 15(a) and 15(b), the X-Limb is able to grasp delicate objects without the need for any complex control algorithms or sensor systems due to the inherent compliance of the hand structure. Fig 15(c) and 15(d) also show that the power and pinch grasp types are capable of grasping spherical objects without the need for an independent grasp type (i.e. without the need for a spherical-grasp). Finally, as shown in Fig 15(e), due to the high gear ration of the motors, in the brake mode of the motors, the power grasp can be used as a hook grip to carry about 10kg.

**Durability test.**    Other than the practical requirements already listed, the durability of the designed prosthetic hand should be considered. Such devices should withstand a minimum number of cycles, maintaining their original functionality. In order to evaluate the durability of the designed soft prosthetic hand, one of the fingers of the X-Limb is connected to the DC motor which is used for actuation of the X-Limb. It underwent 45,000 cycles before some cracks were observed in the membrane of the first and second joint, as shown in Fig 16. Flexure joints, DC motor, sensor, controller and the actuation cables maintained their original functionalities.

According to [38], hand prostheses will typically undergo 120 grasping motions per day. Considering this average daily usage, the designed soft prosthetic hand has an estimated minimum lifetime of one year. In general, there is a trade-off between durability and robustness with weight, size and cost [16]. The trade-off point achieved in the X-Limb design is: weight of 253gr, three grasps types (with capability of individual finger movement), power-grip force of 21.5N, finger flexion speed of 1.3sec, minimum lifetime of one year and a bill of material of around 200 USD (excluding quick disconnect wrist and without the leverage of mass-manufacture).

## Open-source files

In a GitHub repository (https://github.com/MelbourneUniHRL/X-Limb.git), we have provided a library of open-source STL files of the hand design for 3D printing, custom PCB design

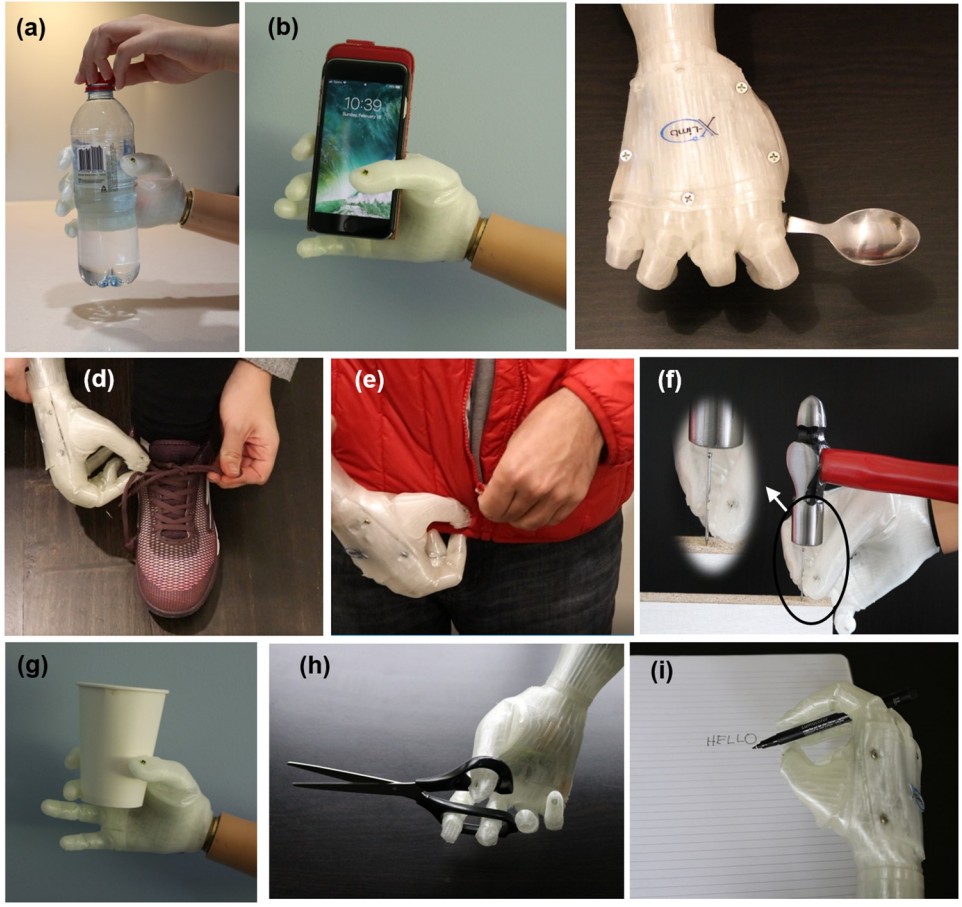

**Fig 14. X-Limb performing three grasp types for carrying out AM-ULA tasks: Power grasp (a,b,c), pinch grasp (d, e,f) and tripod grasp (g,h,i).**

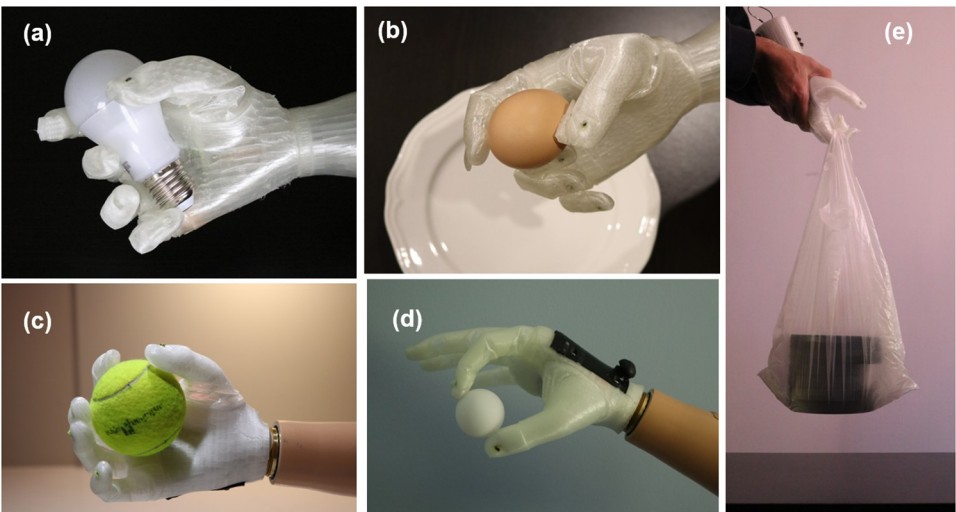

**Fig 15. X-Limb performance in grasping delicate/fragile objects (a and b), spherical objects (c and d) and carrying 10kg weight (e).**

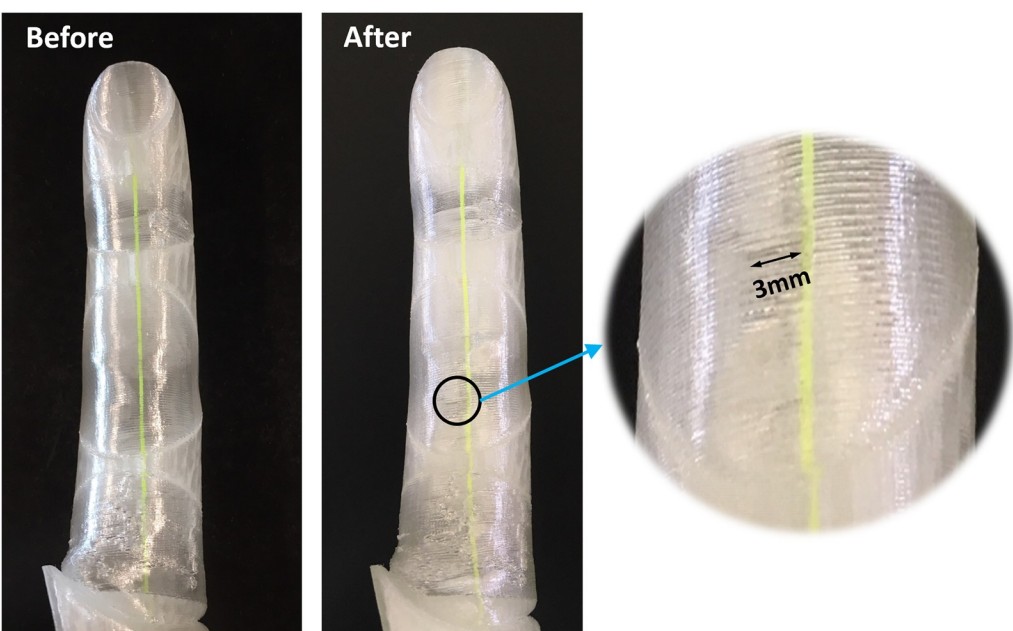

**Fig 16. The test finger appearance before and after 45,000 cycle of flexion and extension of the finger.**

files of the control board, and the hand control source code. The proposed prosthetic hand can be fabricated using the provided open-source files and use as a low-cost alternative for individuals with upper limb loss in developing countries and economically disadvantaged families. These freely accessible files also enables researchers to modify and use it for their own specific applications.

## Conclusion

In this paper, the practical requirement-driven design of a soft robotic prosthetic hand was presented. With design solutions corresponding to the practical requirements, the designed X-Limb was able to satisfy the mechanical/morphological, kinodynamic and functionality requirements. The trade-offs among mechanical/morphological, kinodynamic, and functionality requirements were exploited. With practical requirements factored into the consideration of its design, the X-Limb achieves a weight of 253gr, three grasps types (with capability of individual finger movement), power-grip force of 21.5N, finger flexion speed of 1.3sec, minimum lifetime of one year and cost of 200 USD (excluding quick disconnect wrist). The capability of the X-Limb for practical applications was demonstrated through performing real-world grasping tasks required based on a standard Activities Measure for Upper-Limb Amputees benchmark test.

## Supporting information

**S1 Video. Tendon cable threading.** This video shows threading the tendon cable in the flexure joints with membrane enclosure.
(MP4)

**S2 Video. 3D printing and assembly of X-Limb.** This video shows 3D printing of the X-Limb and assembly of different components.
(MP4)

**S3 Video. Assembly of X-Limb actuation and control system.**
(MP4)

**S4 Video. Performance of X-Limb in carrying out grasping tasks.**
(MP4)

## Author Contributions

**Conceptualization:** Alireza Mohammadi, Denny Oetomo.

**Formal analysis:** Alireza Mohammadi.

**Funding acquisition:** Peter Choong, Denny Oetomo.

**Investigation:** Alireza Mohammadi.

**Methodology:** Alireza Mohammadi, Jim Lavranos, Denny Oetomo.

**Supervision:** Ying Tan, Peter Choong, Denny Oetomo.

**Validation:** Alireza Mohammadi.

**Visualization:** Alireza Mohammadi.

**Writing – original draft:** Alireza Mohammadi, Ying Tan, Denny Oetomo.

**Writing – review & editing:** Alireza Mohammadi, Hao Zhou, Rahim Mutlu, Gursel Alici, Ying Tan, Denny Oetomo.

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
