## [Decision Letter · Decision Letter 0]

31 Mar 2020

PONE-D-20-04803

A practical 3D-printed soft robotic prosthetic hand with multi-articulating capabilities

PLOS ONE

Dear Dr Mohammadi,

Thank you for submitting your manuscript to PLOS ONE. After careful consideration, we feel that it has merit but does not fully meet PLOS ONE’s publication criteria as it currently stands. Therefore, we invite you to submit a revised version of the manuscript that addresses the points raised during the review process.

We would appreciate receiving your revised manuscript by May 15 2020 11:59PM. To enhance the reproducibility of your results, we recommend that if applicable you deposit your laboratory protocols in protocols.io, where a protocol can be assigned its own identifier (DOI) such that it can be cited independently in the future. For instructions see: http://journals.plos.org/plosone/s/submission-guidelines#loc-laboratory-protocols

We look forward to receiving your revised manuscript.

Kind regards,

Luke Connal, PhD

Academic Editor

PLOS ONE

Journal Requirements:

Additional Editor Comments (if provided):

Reviewer #1: Overall the paper is well-written, adequately referenced and provides a sufficient overview of the field of soft, robotic prosthetic hand designs. The design of "X-limb" is well considered in relation to other reported soft prosthetic hands, and the operation and benchmarking has been shown convincingly.

However, the structure and goals of the paper are disjointed, landing somewhere between a review article and a research report. Although this work is submitted as a "collection review", it is not immediately clear that the goal of this work is to synthesise a novel design protocol for a prosthetic hand "X-limb", which is then constructed by the authors and benchmarked. In its current form, the paper reads as a research article with an extended introduction and discussion on design parameters. By combining the remit of a review article and a traditional research article, the paper is difficult to follow and does not meet the format requirements for this journal. That the authors have accumulated the current understanding in prosthetic hand designs to reach the recommendations and benchmarks for "X-limb" is not stated clearly. Instead, it seems like the first half of the paper is a "collection review" and the second half is a research report disclosing the design and performance of "X-limb", which has apparently not been reported previously. In addition, the key design parameters seem to come from just 3 references (15-17), which is inadequate for a review-type article. A major revision of the manuscript is recommended to align the format with either a "collection review" - where the X-limb construction and performance discussion is reduced/removed, and emphasis given to examining the literature - or a "research article" - where the discussion of design elements is given as support for the development of X-limb only, and the construction/performance/benchmarking of this unit is the major element of the work.

Some minor points:

1. The abstract/intro makes no mention of the fact that this manuscript is a "collection review". This should be stated explicitly, and the scope of the literature to be reviewed should be highlighted.

2. The readability of the manuscript overall would benefit from greater use of paragraphs.

3. Include the figure captions alongside the figures.

4. The quantitative data reported for X-limb needs considerably more information provided: what instrument was used to collect this data, how many times were the measurements repeated and how are the units calculated? As a disclosure of novel research, the data needs to form the central component of the paper.

5. More clarity is required in the abstract/intro about exactly what the paper will deliver. In this regard, the conclusion serves as a better example.

Reviewer #2: The authors have successfully constructed soft robotic prosthetic hand using 3D printing. This well-designed soft robot made from polyurethaneis capable of exhibiting multiple functions such as collecting and manipulating objects. This work is systematical and robust, giving high interest to soft robot society. This manuscript is concise, clearly written and well organised. The results are clear and the interpretations are convincing. I recommend the publication of this manuscript in the PLOS ONE. Here are some comments:

1. Why the nozzle temperature was 220ﹾC? How about the glass transition temperature and melting temperature of printed polyurethaneis?

2. The authors claimed that the used polyurethaneis are highly elastic, which should be supported by experimental evidence.

3. Have the authors tried to optimise the 3D printing parameters such as extrusion speed, travel speed and nozzle temperature?
---

## [Author Response · Author response to Decision Letter 0]

18 Apr 2020

A separate file labelled as "Responses To Reviewers" is included in the attached files (appears at the end of document). Here is the text version of that formatted Responses To Reviewers file.

Response to Reviewers for the Manuscript (PONE-D-20-04803)

April 17, 2020

We thank the editors and the reviewers for their valuable and constructive comments and suggestions. We have thoroughly revised the paper according to the reviewers’ comments. The details of the revisions made are provided below, with respect to the comments from the reviewers. The changes in this manuscript compared to the previous have been highlighted in the revised version.

Addressing Comments by Reviewer 1:

Reviewer 1 Summary: Overall the paper is well-written, adequately referenced and provides a sufficient overview of the field of soft, robotic prosthetic hand designs. The design of “X-limb” is well considered in relation to other reported soft prosthetic hands, and the operation and bench- marking has been shown convincingly.

Authors’ Response: We thank the reviewer for recognizing the contributions of this work.

Reviewer Comment 1.1: However, the structure and goals of the paper are disjointed, landing somewhere between a review article and a research report. Although this work is submitted as a “collection review”, it is not immediately clear that the goal of this work is to synthesise a novel design protocol for a prosthetic hand “X-limb”, which is then constructed by the authors and benchmarked. In its current form, the paper reads as a research article with an extended introduction and discussion on design parameters. By combining the remit of a review article and a traditional research article, the paper is difficult to follow and does not meet the format requirements for this journal. That the authors have accumulated the current understanding in prosthetic hand designs to reach the recommendations and benchmarks for “X-limb” is not stated clearly. Instead, it seems like the first half of the paper is a “collection review” and the second half is a research report disclosing the design and performance of “X-limb”, which has apparently not been reported previously. In addition, the key design parameters seem to come from just 3 references (15-17), which is inadequate for a review-type article. A major revision of the manuscript is recommended to align the format with either a “collection review” - where the X-limb construction and performance discussion is reduced/removed, and emphasis given to examining the literature - or a “research article” where the discussion of design elements is given as support for the development of X-limb only, and the construction/performance/benchmarking of this unit is the major element of the work.

Authors’ Response: We thank the reviewer for pointing out this. We believe that the term “Collection Review” for manuscript type had caused the confusion.

Our paper is a research article that presents a systematic design of a practical prosthetic hand. Such a design was driven by identifying the practical requirements, balancing the design complexity and performance with the knowledge of the state-of-the-art technology in the prosthetic hand designs in literature. Hence, our paper is a research article, not a review of the literature.

As to the article type in PLoS One, we were following the instructions given:

- We were invited by the editors of the PLoS Collection on Open Soft Robotics Research (see https://collections.plos.org/s/soft-robotics). As you can see, there is nothing that stipulated that it had to be a review paper. We were invited to submit a paper under the category of ”soft robotic application” – see ”example topics”, last item.

- Under PLoS One ”Criteria for Publication”, (see https://journals.plos.org/plosone/s/criteria- for-publication), it is stated that PLoS One will occasionally commission Collection Reviews or Overviews, but these articles are associated with specific, pre-planned Collections and will not be considered unless solicited. In our case, we were invited by the editors for the Open Soft Robotic Research Collection.

As correctly identified by reviewer, the submitted manuscript presents the design and characterisation of the open-sourced X-Limb as an open-source soft robotic prosthetic hand. This is openly shared with the international research community in a GitHub repository (https://github.com/ MelbourneUniHRL/X-Limb.git). This is also the reason (we believe) that we were invited to submit this paper to this Collection on Open Soft Robotics Research.

We therefore believe that that the manuscript belong squarely in the second category (research article) in its current format. As the Reviewer pointed out, it presents “the discussion of design elements is given as support for the development of X-limb only, and the construction/perfor- mance/benchmarking of this unit as the major element of the work”.

To help clarify this point, we have revised the abstract to reflect the points above in addition to addressing the Reviewer’s other comments below.

Revision made to the manuscript: The Abstract of the paper is revised thoroughly as the following:

“ Soft robotic hands with monolithic structure have shown great potential to be used as prostheses due to their advantages to yield light weight and compact designs as well as its ease of manufacture. However, existing soft prosthetic hands design were often not geared towards addressing some of the practical requirements highlighted in prosthetics research. The gap between the existing designs and the practical requirements significantly hampers the potential to transfer these designs to real-world applications. This work addressed these requirements with the consideration of the trade-off between practicality and performance. These requirements were achieved through exploiting the monolithic 3D printing of soft materials which incorporates membrane enclosed flexure joints in the finger designs, synergy-based thumb motion and cable-driven actuation system in the proposed hand pros- thesis. Our systematic design (tentatively named X-Limb) achieves a weight of 253gr, three grasps types (with capability of individual finger movement), power-grip force of 21.5N, finger flexion speed of 1.3sec, a minimum grasping cycles of 45,000 (while maintaining its original functionality) and a bill of material cost of 200 USD (excluding quick disconnect wrist but without factoring in the cost reduction through mass production). A standard Activities Measure for Upper-Limb Amputees benchmark test was carried out to evaluate the capability of X-Limb in performing grasping task required for activities of daily living. The results show that all the practical design requirements are satisfied, and the proposed soft prosthetic hand is able to perform all the real-world grasping tasks of the benchmark tests, showing great potential in improving life quality of individuals with upper limb loss.”

Some minor points:

Reviewer Comment 1.2 The abstract/intro makes no mention of the fact that this manuscript is a ”collection review”. This should be stated explicitly, and the scope of the literature to be reviewed should be highlighted.

Authors’ Response: As we explained in response to the previous comment, this manuscript is a research article. It is however submitted to the Collection on Soft Robotics Research. We will consult the editor on the appropriate way to highlight this in the abstract / introduction.

Reviewer Comment 1.3 The readability of the manuscript overall would benefit from greater use of paragraphs.

Authors’ Response: The manuscript has been revised thoroughly to improve the readability. Paragraphs have been further organised, especially in the Introduction, to allow better presentations of the logical points.

Reviewer Comment 1.4 Include the figure captions alongside the figures.

Authors’ Response: This is the required submission structure provided by PLoS One template. The final manuscript will be re-arranged by the editorial team of PLoS.

Reviewer Comment 1.5 The quantitative data reported for X-limb needs considerably more information provided: what instrument was used to collect this data; how many times were the measurements repeated and how are the units calculated? As a disclosure of novel research, the data needs to form the central component of the paper.

Authors’ Response: We thank the reviewer for pointing out this. In the following table (Table 1 below), we have listed all the quantitative data used in the manuscript and explained the information that has been added to the manuscript. The ones in red has been added to the manuscript.

Reviewer Comment 1.6 More clarity is required in the abstract/intro about exactly what the paper will deliver. In this regard, the conclusion serves as a better example.

Authors’ Response: The Abstract has been revised thoroughly to address the comment of the reviewer.

Table 1: List of quantitative data reported for X-Limb

Quantitative data: What instrument or method is used / Units / Number of iterations

1) Weight: “digital scale” has been added to Page 10, Line 378 / gr (as stated in the first row of the Table 3)

2) Number of Degrees of Freedom (DoF): “In the X-Limb design, each of index, middle, ring, and little fingers has three joints (distal, proximal , and metacarpal) as shown in Fig. 1 and Fig. 2 and thumb has one metacarpal joint as shown in Fig. 3 which results in 13 DoFs of the X-Limb. One actuator is required to actuate each of these underactuated fingers.” has been added to Page 9, Line 309

3) Number of actuators: “actuation system that consists of five geared DC micromotors (6V HPCB Micro Metal Gearmotor, Pololu Inc.), see Fig. 5. Each of these motors actuates one of the fingers. ” as stated in Page 9, Line 314

4) Pinch and power grip force: The instrument for measuring pinch and power grip force is shown in Fig. 8 / The unit is N (Newton) as stated in the first two the Table 3 in the manuscript / “The maximum pinch grip force (before the index finger loses contact with the thumb) was measured to be 10.2±0.4N at 50 iterations of the force measurement. The power grip force was measured using a cylindrical object with diameter of 40mm and two FSR sensors placed in the middle (Fig. 8(b)). This experiment is also iterated for 50 times which resulted in power grip force of 21.5±1.2N at full flexion of the fingers.” has been added to the manuscript in Page 11, Line 401 regarding the number of iterations and the results

5) Grasp speed: “ The hand grasp speed is defined as the finger flexion speed or the time required for full flexion of the hand. For the X-Limb, the time required for motor spool to pull the tendon cable from full extension to full flexion of the middle finger (as the slowest finger) is obtained using the position sensor attache to the motor shaft. The obtained time is 1.3sec which satisfies the design requirement for the hand closing speed.” has been added to the manuscript in Page 11, Line 387 to clarify the method used for measuring the grasp speed.

6) Torsional stiffness: The experimental setup for measuring the torque and torsional angle is shown

in Fig. 9. The torque and angle measurements are explained in the manuscript in Page 12, Line 442 “The finger is actuated in 2 degree increments and the quasi-static torque required for the case of the joint with membrane and without membrane were measured. ”

7) Flexural stiffness: The experimental setup for measuring the flexure joint in shown in Fig. 11. The tendon cable force measurement has been added to the manuscript in Page 13, Line 461 “The tendon cable force is measured using the load cell attached to the force testing machine.” The method of measuring the flexion angle has been added to the manuscript in Page 13, Line 463 “ The sequential images of the joint flexion has been processed in SolidWorks (Dassault Systems Inc.) to obtain the flexion angle at each applied force measurement.”

8) Life cycle of a single finger: “In order to evaluate the durability of the designed soft prosthetic hand, one of the fingers of the X-Limb is connected to the DC motor which is used for actuation of the X-Limb.” in Page 14, Line 513 of the manuscript states the instrument used for obtaining the life cycle of a single finger.

Addressing Comments by Reviewer 2

Reviewer 2 Summary: The authors have successfully constructed soft robotic prosthetic hand using 3D printing. This well-designed soft robot made from polyurethaneis capable of exhibiting multiple functions such as collecting and manipulating objects. This work is systematical and robust, giving high interest to soft robot society. This manuscript is concise, clearly written and well organised. The results are clear and the interpretations are convincing. I recommend the publication of this manuscript in the PLOS ONE.

Authors’ Response: We thank the reviewer for recognizing the contributions of this work.

Here are some comments:

Reviewer Comment 1.1 Why the nozzle temperature was 220◦C? How about the glass transition temperature and melting temperature of printed polyurethaneis?

Authors’ Response: The material of the filament used for 3D printing of the proposed soft robotic hand is Thermoplastic Polyurethane (TPU) and the nozzle temperature of 220◦C is the recommended temperature by the filament manufacturer for melting and extruding of the filament during 3D printing.

The glass transition temperature and melting temperature of TPU is -44◦C and 195◦C, respec- tively, as stated in the following reference:

Chuayjuljit, S. and Ketthongmongkol, S., 2013. Properties and morphology of injection-and compression-molded thermoplastic polyurethane/polypropylene-graft-maleic anhydride/wollastonite composites. Journal of Thermoplastic Composite Materials, 26(7), pp.923-935.

Therefore, the glass transition temperature of TPU is below the room temperature.

Revision made to the manuscript: The following has been added to the manuscript in Page 6, Line 187:

“The nozzle temperature was set to 220oC which is the recommended value by TPU filament (Fi-

laForm) manufacturer, considering the glass transition temperature and melting temperature of TPU at -44◦C and 195◦C, respectively. ”

Reviewer Comment 1.2 The authors claimed that the used polyurethaneis are highly elastic, which should be supported by experimental evidence.

Authors’ Response: We thank the reviewer for pointing out this. We have included the following reference for the hyper-elastic behaviour of the TPU filament and the reference in given in the manuscript in Page 6, Line 175:

Qi, Hang Jerry, and Mary C. Boyce. ”Stress–strain behavior of thermoplastic polyurethanes.” Mechanics of Materials 37.8 (2005): 817-839.

Reviewer Comment 1.3 Have the authors tried to optimise the 3D printing parameters such as extrusion speed, travel speed and nozzle temperature?

Authors’ Response: The 3D printer used in this paper (FlashForge) has an open-source software (FlashPrint) for slicing the STL files and preparing the file for 3D printing. In this software, by choosing the printing material, the extrusion speed and travel speed will be set automatically to an appropriate values. We have not attempted optimisation to the printing parameters. In our case “Flexible Filament” is used as Material Type. The nozzle temperature, as mentioned in the response to the Comment 2.1, is provided by the filament manufacturer.

Revision made to the manuscript: The following has been added to the manuscript in Page 6, Line 182:

“ The 3D printer used in this paper (FlashForge) has an open-source software (FlashPrint) for slicing the STL files and preparing the file for 3D printing. In this software, by choosing the printing material, the extrusion speed and travel speed will be set automatically to an appropriate values. In our case “Flexible Filament” is used as Material Type which results in the extrusion speed of 60mm/s and travel speed of 80mm/s. ”

---

## [Editor Report · Decision Letter 1]

22 Apr 2020

A practical 3D-printed soft robotic prosthetic hand with multi-articulating capabilities

PONE-D-20-04803R1

Dear Dr. Mohammadi,

We are pleased to inform you that your manuscript has been judged scientifically suitable for publication and will be formally accepted for publication once it complies with all outstanding technical requirements.

With kind regards,

Luke Connal, PhD

Academic Editor

PLOS ONE
---

## [Editor Report · Acceptance letter]

1 May 2020

PONE-D-20-04803R1 

A practical 3D-printed soft robotic prosthetic hand with multi-articulating capabilities 

Dear Dr. Mohammadi:

I am pleased to inform you that your manuscript has been deemed suitable for publication in PLOS ONE. Congratulations! Your manuscript is now with our production department. 

With kind regards,

on behalf of

Dr. Luke Connal 

Academic Editor

PLOS ONE